

# Large-scale movements of common bottlenose dolphins in the Atlantic: dolphins with an international courtyard

Ana Dinis[1,2], Carlota Molina[2,3], Marta Tobeña[4], Annalisa Sambolino[1,2], Karin Hartman[5], Marc Fernandez[1,6], Sara Magalhães[7], Rui Peres dos Santos[8], Fabian Ritter[9], Vidal Martín[10], Natacha Aguilar de Soto[11] and Filipe Alves[1,2]

[1] Mare-Marine and Environmental Sciences Centre, Agência Regional para o Desenvolvimento da Investigação Tecnologia e Inovação (ARDITI), Funchal, Madeira, Portugal
[2] OOM - Oceanic Observatory of Madeira, Funchal, Madeira, Portugal
[3] Department of Animal Biology, Ecology and Environmental Sciences, University of Barcelona, Barcelona, Catalonia, Spain
[4] Centro I&D Okeanos, University of Azores, Horta, Azores, Portugal
[5] Risso's Dolphin Research Center, Nova Atlantis Foundation, Pico, Azores, Portugal
[6] cE3c - Centre for Ecology, Evolution and Environmental Changes/Azorean Biodiversity Group, University of Azores, Ponta Delgada, Azores, Portugal
[7] Mar Ilimitado, Sagres, Portugal
[8] Futurismo, Pico, Azores, Portugal
[9] M.E.E.R. e.V, Berlin, Germany
[10] SECAC Society for the Study of Cetaceans in the Canary Archipelago, Lanzarote, Canary Island, Spain
[11] BIOECOMAC, Department of Animal Biology, University of La Laguna, Tenerife, Canary Island, Spain

Corresponding author
Ana Dinis, ana.dinis@mare-centre.pt

## ABSTRACT

Wide-ranging connectivity patterns of common bottlenose dolphins (*Tursiops truncatus*) are generally poorly known worldwide and more so within the oceanic archipelagos of Macaronesia in the North East (NE) Atlantic. This study aimed to identify long-range movements between the archipelagos of Macaronesia that lie between 500 and 1,500 km apart, and between Madeira archipelago and the Portuguese continental shelf, through the compilation and comparison of bottlenose dolphin's photo-identification catalogues from different regions: one from Madeira ($n = 363$ individuals), two from different areas in the Azores ($n = 495$ and 176), and four from different islands of the Canary Islands ($n = 182$, 110, 142 and 281), summing up 1791 photographs. An additional comparison was made between the Madeira catalogue and one catalogue from Sagres, on the southwest tip of the Iberian Peninsula ($n = 359$). Results showed 26 individual matches, mostly between Madeira and the Canary Islands ($n = 23$), and between Azores and Madeira ($n = 3$). No matches were found between the Canary Islands and the Azores, nor between Madeira and Sagres. There were no individuals identified in all three archipelagos. The minimum time recorded between sightings in two different archipelagos ($\approx 460$ km apart) was 62 days. Association patterns revealed that the individuals moving between archipelagos were connected to resident, migrant and transient individuals in Madeira. The higher number of individuals that were re-sighted between Madeira and the Canary Islands can be explained by the relative proximity of these two archipelagos. This study shows the first inter-archipelago movements of bottlenose dolphins in the Macaronesia region, emphasizing the high mobility of this species and supporting the high gene flow described for oceanic

dolphins inhabiting the North Atlantic. The dynamics of these long-range movements strongly denotes the need to review marine protected areas established for this species in each archipelago, calling for joint resolutions from three autonomous regions belonging to two EU countries.

# INTRODUCTION

The common bottlenose dolphin *Tursiops truncatus*, (hereafter "bottlenose dolphin"), like other cetaceans, faces a variety of anthropogenic disturbances, such as water pollution, incidental capture (by-catch) or vessel collisions (*Wells & Scott, 2018*). Coastal and pelagic variations or ecotypes of bottlenose dolphins have been described based on morphological, ecological and genetic differences (*Oudejans et al., 2015*). The well-studied populations of coastal bottlenose dolphins exhibit a variety of horizontal movements, including seasonal migrations, year-around home ranges, periodic residency, and a combination of occasional long-range movements and repeated local residency (*Shane, Wells & Würsig, 1986*; *Wells & Scott, 2018*). However, much less is known about the ranging patterns of pelagic bottlenose dolphins (*Wells & Scott, 2018*). It is crucial to gain a better understanding of the ranging patterns of this species in order to establish suitable conservation measures. Apart from small scale movements of bottlenose dolphin studied in greater depth (e.g., *Reynolds, Wells & Eide, 2000*; *Silva et al., 2008*; *Tobeña et al., 2014*; *Hwang et al., 2014*; *Dinis et al., 2016*), information from long-distance and inter-archipelagos movements is scarce. Insufficient information on long-distance movements may result in higher emphasis on residency (*Bearzi, Bonizzoni & Gonzalvo, 2011*), when in fact individuals may leave the study area more frequently than initially thought. Previous studies of pelagic bottlenose dolphin populations in the NE Atlantic area suggested that these populations have a high gene flow and are genetically less differentiated (*Querouil et al., 2007*; *Louis et al., 2014*). Additionally, different residency patterns and individual movements within each archipelago were identified for the Azores (*Silva et al., 2008*), the Canary Islands (*Tobeña et al., 2014*) and Madeira (*Dinis et al., 2016*), with just a portion of the individuals being classified as residents. These results indicate large individual home ranges, but there is no evidence of the connectivity of the populations between these oceanic archipelagos. A recent photo-identification study demonstrated the connectivity of pilot whales within the Macaronesia biogeographical region (*Alves et al., 2018a*; *Alves et al., 2019*), also highlighting the importance of such studies for conservation. Hence, it can be speculated that other highly mobile species like bottlenose dolphin can also perform long-range movements in this region (*Silva et al., 2008*; *Dinis et al., 2016*). We investigated for the first time horizontal large-scale movements of this species between the archipelagos of Madeira, Azores and the Canary Islands, i.e., within the biogeographical region of Macaronesia, and with the Portuguese continental shelf, covering an area of more than 1,600,000 km2. The present study aims to the better understanding of the bottlenose dolphin connectivity among
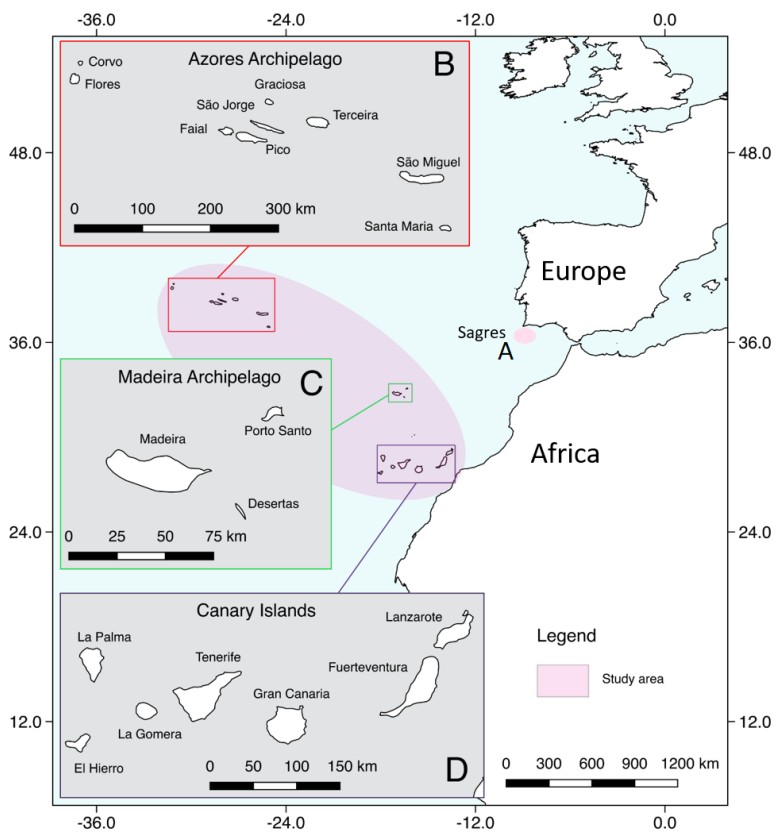

**Figure 1** **Map showing the study area.** (A) Sagres, (B) Azores, (C) Madeira, (D) Canary Islands (extracted from Natural Earth: https://www.naturalearthdata.com/).

these remote oceanic archipelagos, and to help in this species' future conservation and management efforts.

## MATERIALS & METHODS

### Study area

The study area included the oceanic archipelagos of Madeira, Azores and the Canary Islands in the Macaronesia region, plus an adjacent coastal area along the Iberian Peninsula (Fig. 1). Macaronesia consists of island archipelagos located in the Northeast Atlantic Ocean, off the coasts of Europe and West Africa (*Almada et al., 2013*). It has a unique marine fauna, which has been influenced by West Africa, the Mediterranean Sea and continental western Europe (*Floeter et al., 2008*; *Almada et al., 2013*), making this region an ideal habitat for a high number of cetacean species (*Pérez-Vallazza et al., 2008*; *Freitas et al., 2012*; *Silva et al., 2014*; *Alves et al., 2018b*).

### Photo-identification data

Dolphin movements were determined through the cross-comparison of photo-identification catalogues held by eight organizations in Portugal and Spain (Table 1). The Madeira catalogue was compiled by Oceanic Observatory of Madeira (OOM) and

**Table 1   Summary of the photo-identification data used in this study.**

| Number of individual dolphins | Source | Period | Location |
|---|---|---|---|
| 363 | Oceanic Observatory of Madeira (OOM) | 2004–2016 | Madeira island |
| 176 | Nova Atlantis Foundation | 2003-2007 | Pico (Azores) |
| 495 | MONICET-University of Azores | 2004–2016 | Pico, Faial, São Miguel and Terceira (Azores) |
| 42 | Espaço Thalassa | 2014–2016 | Pico and Faial (Azores) |
| 182 | SECAC | 2004–2015 | La Gomera (Canary Islands) |
| 110 | SECAC | 2014 | Tenerife (Canary Islands) |
| 142 | SECAC | 2010–2011 | La Palma (Canary islands) |
| 281 | BIOECOMAC-University of La Laguna/NGO M.E.E.R. e.V. | 2001–2011 | La Palma, La Gomera and Tenerife (Canary islands) |
| 359 | Mar Ilimitado | 2007–2015 | Sagres |

comprised 363 individuals collected between 2004 and 2016, and sighted mainly off the south coast of Madeira. Two catalogues from the Azores were included, one containing 176 individuals from Pico and Faial islands collected between 2003 to 2007 compiled by Nova Atlantis Foundation, and a second one with 495 individuals from Pico, Faial, São Miguel and Terceira islands, collected between 2004–2016 compiled a through a long-term citizen science program focused on whale-wacthing touristic operations in the Azores, called MONICET (MONItoring CETaceans). A third set of raw data from the Azores (Pico and Faial islands), containing 201 photos, from which 42 individuals were identified by OOM, collected by a whale-watching company (Espaco Thalassa), between 2014 and 2016 was added. From the Canary Islands, four catalogues from two institutions and from different islands were used: one from La Gomera with 182 individuals (2004–2015); one from Tenerife with 110 individuals (2014); one from La Palma with 142 individuals (2010–2011 and 2015), all compiled by SECAC (Sociedad para el Estudio de los Cetáceos en el Archipiélago Canario), and one with 281 individuals (2001–2011), that included photos from La Gomera, El Hierro and La Palma, compiled by BIOECOMAC (Biodiversidad, Ecología marina y Conservación de la Universidad de La Laguna), using their own data and data from a local NGO called M.E.E.R. e.V.(Mammals, Encounters, Education and Research - La Gomera). The catalogue from Sagres contained 359 individual photographed from 2001 until 2016 and was compiled by the whale-watching company Mar Ilimitado.

The catalogues used, were built using different sources, ranging from whale watching operators to research teams and independent photographers and were constructed by creating a dataset of capture histories, using individual information taken by photographs (following *Würsig & Jefferson, 1990*). Photographs were graded according to their level of focus, contrast, exposure and angle of the dorsal fin; and level of distinctiveness of the individuals was graded according to the number of nicks and notches present in the dorsal fin. Only good quality photos and distinct and very distinct individuals were used in this analysis in order to enhance the reliability of the matches (*Urian et al., 2015*). Whenever a match was found and confirmed, the same identification number as that of

the individual stored in the database was assigned, but, if there were no match, a new identification number was attributed to that individual and it was added to the catalogue as a new individual (*Dinis et al., 2016*). The matching procedure was conducted through the comparison of natural markings like nicks and notches on the dorsal fin, and the shape of the fin (*Würsig & Würsig, 1977*). In all the catalogues, with the exception the one made by BIOECOMAC, the comparison was conducted by the same researcher by naked eye, and confirmed by a second experienced researcher. If doubts persisted, a third experienced researcher would double-check. In the catalogue compiled by BIOECOMAC, dorsal fin images were entered into a digital database using the software Darwin 2.0 (©Eckerd College Dolphin Research Group), a trailing edge contour was extracted, which was identifiable from both sides (*Auger-Méthé & Whitehead, 2007*), and the software was used to assist the matching of individual dolphins (*Tobeña et al., 2014*).

## Macaronesia individuals: photo-identification analysis

The Macaronesia database, containing only the individual matches, was compiled by comparing the individual catalogues introduced in the previous section. The comparison was made following the procedures described above, by naked eye, always by the same researcher. The researcher graded all photographs according to their level of distinctiveness and quality, only using photographs with good quality and individuals that were distinct and very distinct. When a match was found, an identification code (the Macaronesia identification code) was created, for that individual both pictures of the dolphin were added to the database and both locations were indicated in the capture history dataset. Only dolphins seen in two or more archipelagos and matches with 100% certainty, when confirmed by a second experience researcher, were included in this database.

## Associations and residency in Madeira archipelago

The study of the association patterns was made for Madeira archipelago data, including the individuals that were seen in more than one archipelago. It aimed to investigate the residency pattern of these individuals in Madeira and their connectedness with the other dolphins identified in this archipelago. Individuals from the Madeira catalogue, seen in association with other individuals between 2004 and 2016 were used in this analysis. Associations between individuals were analyzed according to residency patterns established for this archipelago (*Dinis et al., 2016*). Residency patterns were assigned to individual dolphins based on their capture histories. The term 'resident' was used to designate dolphins that were seen regularly during the study period in the study area (during three seasons in a year and in more than two consecutive years), 'transient' dolphins were defined as those seen just once in the main area and dolphins seen more than once, but in non-consecutive years, were considered 'migrants'. A social network diagram was created using NetDraw 2.160 (*Borgatti, 2002*) to visualize individual association.

# RESULTS

## Photo-identification analysis

There were 26 dolphins with matches: 23 between Madeira and Canary Islands ($\approx$500 km apart), and three individuals between the Azores and Madeira ($\approx$ 1,000 km apart). No

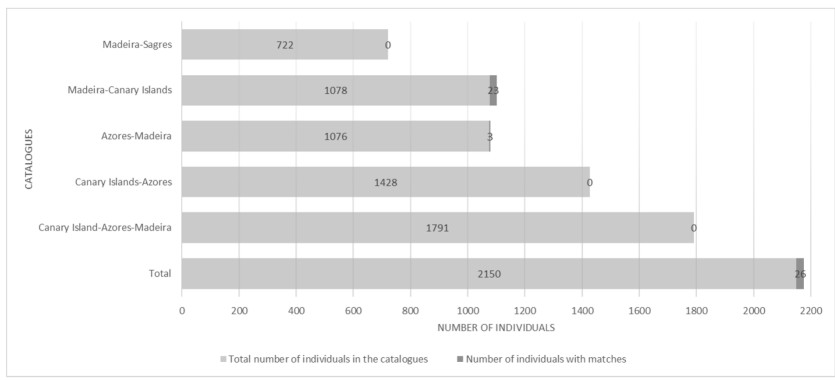

**Figure 2** Number of individuals in the catalogues and number of individuals with matches, distributed by areas.

matches between the Canary Islands and the Azores were found. Likewise, none of the individuals were seen in all three archipelagos, nor between Madeira and Sagres (Fig. 2). The 23 matches between Madeira and the Canary Islands (occurred on three of the four studied islands in the Canary Islands, mainly with El Hierro ($n = 6$, ≈570 km) and La Palma ($n = 14$, ≈460 km) (Table S1). The results also showed back and forth movements made by Tt_MAC_8 and Tt_MAC_12, between Madeira and the Canary Islands, representing a round-trip of approximately 920 km (Fig. 3). Moreover, two individuals were seen within the Canary Islands, and then off Madeira several years later: Tt_MAC_3 was sighted seven times intermittently off El Hierro in 2004, 2008, 2009, then was photographed off La Palma in 2010, and sighted two times off Madeira in 2014 and in 2016. Tt_MAC_4 was first seen off El Hierro in 2009, then sighted off the neighboring island of La Gomera in 2010, was observed again in El Hierro in 2010 and 2011, and eventually sighted off Madeira in 2015 (Table S1). Four individuals (Tt_MAC_7, 11, 13 and 17) were sighted off La Palma on the same date (on 24th May 2011) and then sighted together off Madeira on 13th August 2011 with less than 3 months between re-sightings (Fig. 4). Tt_MAC_9, 12, 14 and 15 were sighted in the same time frame and in the same locations (Table S1).

The three individuals seen first in the Azores and last off Madeira were sighted three (Tt_MAC_24), nine (Tt_MAC_25) and 10 (Tt_MAC_26) years apart. Tt_MAC_24 was seen in Pico island, which represents a distance to Madeira of approximately 1,200 km, while Tt_MAC_25 and 26 were sighted off São Miguel which represents a distance to Madeira of roughly 950 km. No movements from Madeira to Azores were recorded (Fig. 5).

Tt_MAC_17 was photographed off La Palma and then off Madeira within 62 days, presenting the minimum time interval that an individual travelled between two archipelagos, covering around 460 km within this timeframe.

## Associations and residency in Madeira Archipelago

The social network diagram (Fig. 6) incorporated 332 individual dolphins, catalogued in Madeira archipelago, and presents three clusters grouped by residency patterns. Seventeen dolphins were seen both in the Canary Islands and in Madeira associated with all categories

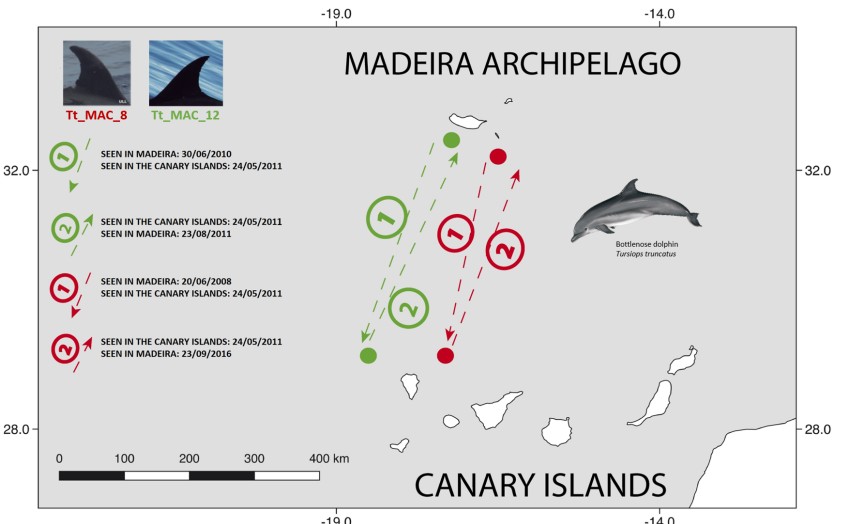

**Figure 3** **Map showing the two-way movements of two bottlenose dolphins between Madeira Island and La Palma, in the Canary Islands (round-trip of ≈ 920 km).** The dots are figurative and do not reflect the exact location of the dolphins. Illustration by E. Berninsone © ARDITI.

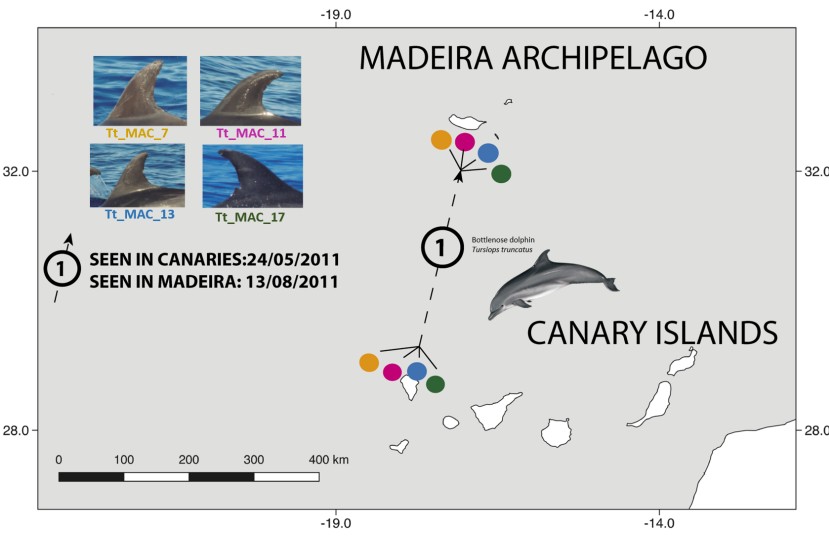

**Figure 4** **Map showing the movement of four bottlenose dolphins between the island of La Palma, in the Canary Islands and Madeira (≈ 500 km).** The dots are figurative and do not reflect the exact location of the dolphins. Illustration by E. Berninsone © ARDITI.

of residency patterns. Two dolphins seen both in Azores and Madeira (Tt_MAC_24 and 25) associated with migrant individuals seen both in Madeira and in the Canary Island (Tt_MAC_3 and 20), and the third (Tt_MAC_26) was seen in association with transient dolphins.

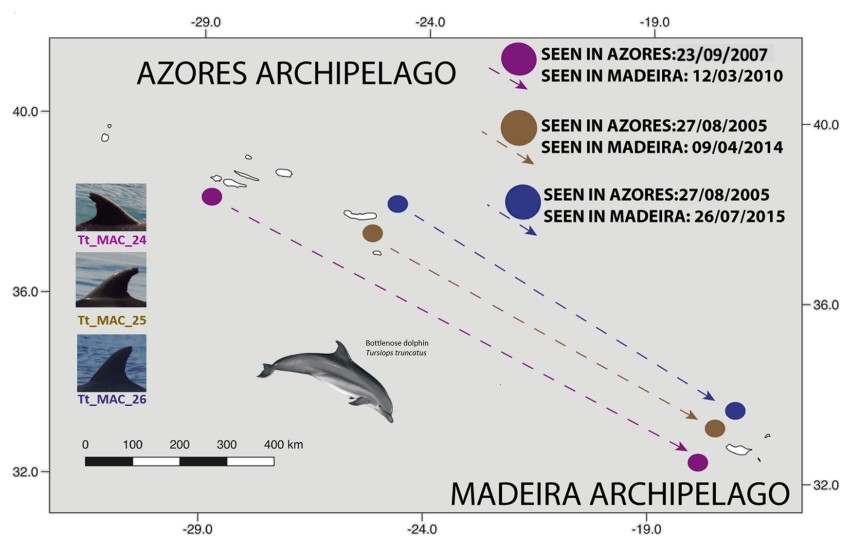

**Figure 5** **Map showing the movement of three bottlenose dolphins between the Azores (Pico and São Miguel islands), and Madeira archipelagos (≈ 1,000 km).** The dots are figurative and do not reflect the exact location of the dolphins. Illustration by E. Berninsone © ARDITI.

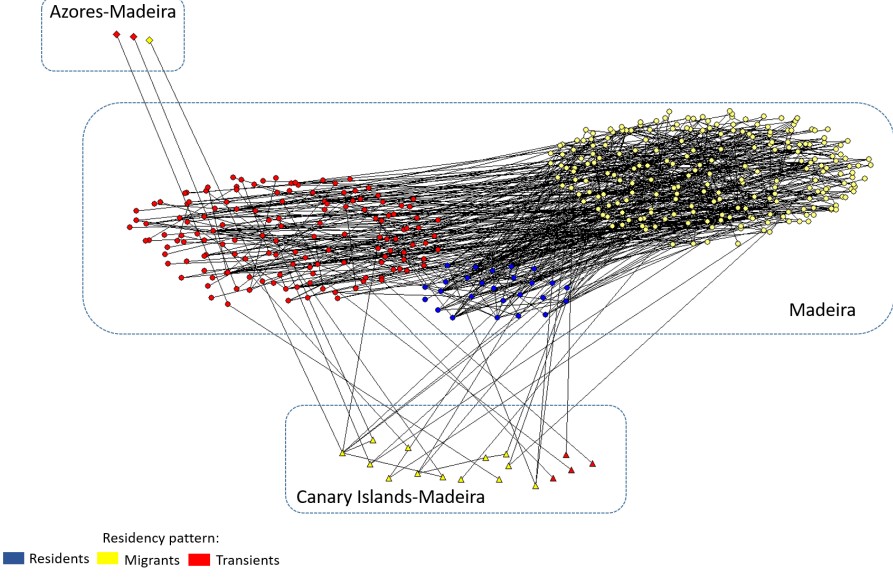

**Figure 6** **Social network diagram illustrating the associations between the dolphins with different residency patterns identified in Madeira, and the 20 dolphins seen in association in more than one archipelago.** Individual dolphins are represented by nodes and associations by the lines between nodes. Nodes color and shape indicates the archipelago of capture and residency pattern in Madeira archipelago.

## DISCUSSION

This study shows that 26 bottlenose dolphins photo-identified off Madeira moved between Macaronesian archipelagos, demonstrating that this species' population covers wide areas in the NE Atlantic. These 26 individuals correspond to 7.1% of the 363 catalogued dolphins in the Madeira archipelago, similarly to what was found for UK and Irish waters (approximately 6%, *Robinson et al., 2012*). Only a few studies described long-distance movements (>1,000 km) of bottlenose dolphin around the world (e.g., *Wood, 1998*; *Wells et al., 1999*; *Robinson et al., 2012*), and none covered these three archipelagos of the Macaronesia region so far, thus this study expands our knowledge of the species in this area of the NE Atlantic. Previous examples of wider-scale movements based on photo-identified bottlenose dolphins come from Argentina (*Würsig, 1978*), Ireland (*O'Brien et al., 2009*), Mediterranean Sea (*Gnone et al., 2011*), and eastern North Pacific Ocean (*Defran et al., 2006*; *Hwang et al., 2014*). For example, off Argentina, one individual travelled 300 km, while off the coast of Ireland, an individual travelled a distance as large as 650 km. The distances reported here for the individuals that moved between Madeira and the Canary Islands are comparable to these ones, and if we consider the round-trip, the distance travelled is even larger, similar to the 965 km covered by a dolphin that travelled from Mexico to the USA, described by *Hwang et al. (2014)*. The distance travelled by Tt_MAC_24 seen off Pico island as well as off Madeira Island, represents a distance of approximately 1,200 km, one of the highest distances recorded so far for this species. It comes closer to the 1277 km an individual travelled between UK and Ireland (*Robinson et al., 2012*).

The inshore waters of the oceanic archipelagos within the NE Atlantic waters offer a sheltered place where bottlenose dolphins can feed, when compared to the offshore waters nearby (*Silva et al., 2008*; *Dinis et al., 2016*). Possibly, when food resources are scarce, some individuals may travel longer distances to where similar, and more abundant food resources may be available. In less productive habitats such as oceanic waters, animals can be expected to have larger home ranges because there is a need to range further to find sufficient food (*Silva et al., 2008*; *Bräger & Bräger, 2019*).

The back and forth movements we found demonstrate that at least some of the bottlenose dolphins in Macaronesia have very large home ranges that include more than one archipelago. One would expect that the dolphins prefer to travel comparably shorter distances because it would imply less effort. This might explain the higher number of matches between the Madeira archipelago and the Canary Islands as compared to the greater distance between the Azores and the Canaries. In addition, Madeira archipelago and the Canary Islands share many biogeographic, and likely also oceanographic characteristics. Freitas and colleagues (*2019*) speculate that Madeira and the Canary Islands should constitute a formal biogeographic unit when referring to the high number of shared endemic marine species. The same study affirms that genetic interchange (e.g., larvae dispersion, colonization events) occur much more frequently between these two archipelagos than with other areas of Macaronesia.

Although we could not determine the sex of the dolphins seen in more than one archipelago, for male bottlenose dolphins, long-distance movements could also serve to

get access to receptive females outside their own population. I.e., young adult males could be driven to seek for females, as described for Indo-pacific bottlenose dolphin (*Tursiops aduncus*) in Shark Bay, Australia (*Connor, Smolker & Richards, 1992*), and thereby also increasing gene flow between populations. In this way, population viability could be improved and genetic differences within the NE Atlantic bottlenose dolphin populations may perhaps decrease, as confirmed by a study that compared individuals from the Madeira and Azores archipelagos (*Querouil et al., 2007*).

*Tobeña et al. (2014)*, in a study reporting inter-islands movements within the Canary Islands, described two individuals that were seen over a long period of time (three and four years). These two individuals are Tt_MAC_3 and 4 in this study, suggesting that even individuals that were considered resident in an area or having a high degree of site fidelity may undertake long-range movements from time to time. Another cross-Macaronesian study (*Alves et al., 2018a*) reported a group of five socially related short-finned pilot whales with strong site fidelity to Madeira which made a round trip to the Azores archipelago, covering approximately 2,000 km, highlighting the importance of caution when assigning residency patterns to smaller areas in oceanic waters. Similarly, in the study of long-range movements of bottlenose dolphins (*Robinson et al., 2012*), the far ranging individuals had been considered to belong to discrete resident populations in the UK and Ireland.

Four individuals (Tt_MAC 7, 11, 13, 17) were seen together off La Palma and were encountered thereafter in Madeira (Fig. 4). Our results also showed that other Macaronesian individuals (Tt_MAC_9, 12, 14 and 15) were documented during the same period in both archipelagos, indicating stable social association, which may persist during, or even favor, long-range oceanic journeys.

Bottlenose dolphins' social structure vary between locations, and even individuals from the same community may behave differently (*Gowans, 2019*; *Genov et al., 2019*). Our network analysis for the Madeira archipelago revealed that the Macaronesian bottlenose dolphins were seen with transients, migrants and resident dolphins, including one resident that has a high level of centrality (*Dinis et al., 2016*). This indicates that some far-ranging dolphins are connected to individuals that play a central role for connectivity of local network as social brokers (*Lusseau & Newman, 2004*). Individuals exhibiting extended home ranges can have a fundamental role, contributing to a genetic variability in oceanic dolphin communities, which otherwise would be genetically isolated (*Louis et al., 2014*).

The minimum period of time between the re-captures in different archipelagos (Canary Islands to Madeira) was 62 days. Satellite-monitored movements of an individual bottlenose dolphin off Florida showed that the dolphin moved 581 km in 25 days (*Mate et al., 1995*). In Japan, one tagged bottlenose dolphin travelled about 604 km in 18 days (*Tanaka, 1987*). Therefore, the time period documented in this study is comparatively long, but the actual time it took the dolphins to cover the distance from one archipelago to the other remains unknown. In one study using satellite telemetry (*Klatsky, Wells & Sweeney, 2007*), the authors determined a mean travel distance of 28.3 km/day for three offshore bottlenose dolphins, which suggests that the dolphins reported here could have covered the distance within a time period well below 62 days. Alternatively, they may also have travelled a much longer distance within those 2 months.

The fact that we did not find any match between the Madeira archipelago and the Portuguese continental shelf should not exclude the assumption that some individuals may undertake these even longer trips. A previous study on bottlenose dolphin populations of the NE Atlantic (*Louis et al., 2014*) found no genetic structure between the Azores archipelago and individuals from several parts of the NE Atlantic, including the shelf-edge.

Connectivity studies can be a monitoring tool when assessing ranging patterns over wider areas, as has been regularly made for large whales (e.g., *Robbins et al., 2011*; *Bertulli, Rasmussen & Tetley, 2013*; *Carpinelli et al., 2014*). We now know that at least some bottlenose dolphins perform extreme mobility throughout the Macaronesia region. This has multiple implications for conservation and management efforts designed for this species: Firstly, management units may not be separable and their connectivity must be taken into account e.g., when establishing marine protected areas (MPAs). Connected populations will have to be considered coherently within conservation frameworks such as the European Union Habitats & Species Directive (HD). Bottlenose dolphins inhabiting Macaronesia waters are, as in other places, subject to many threats like fisheries interaction (by-catch), overfishing, pollution, vessel strikes, stress caused by human recreational activities such as whale-watching and climate change, among others (*Reeves, 2018*). In the Macaronesia region a large number of marine protected areas were designed to protect bottlenose dolphins, but with different levels of protection (*Hoyt, 2011*). Some of these are SACs (Special Area of Conservation) designated as part of the Natura 2000 network under the European Union HD. Most marine SACs thereby only cover coastal areas, rather than reaching offshore. While the establishment of MPAs is a step forward to protect bottlenose dolphins (*Hoyt, 2011*; *Silva et al., 2012*) in this region, more has to be done in terms of mitigations measures, as many of the established SACs still lack management plans. In the Azores, it has been demonstrated that the established areas are not sufficient mainly because they are not covering the complete home range of the dolphins (*Silva et al., 2012*). The same applies to the Canary Islands and to Madeira archipelago. Our results confirm that the bottlenose dolphins' home range in Macaronesia includes more than one archipelago and the offshore waters around them. This means that SACs should be expanded to include offshore waters allowing protection measures to be more effective. Such an expansion would have positive side effect for other highly mobile species, like the short-finned pilot whale, that are known to use this area widely, too (*Alves et al., 2019*).

## CONCLUSIONS

This study provides first evidence of large-scale connectivity of bottlenose dolphin communities between Macaronesia archipelagos, highlighting the strength of combining photo-identification catalogues from different areas, and can be seen as a first step to review the established boundaries of the existing MPAs (SACs) for this species in Macaronesia. This will require a considerable effort, because there are three different autonomous communities (Madeira, Azores and Canary Islands) involved, belonging to two EU member states (Portugal and Spain). Nevertheless, it would correspond to an adaptive and ecosystem-based management approach and serve the coherent protection of the species across borders —all aspects that the EU HD strives to achieve.

## ACKNOWLEDGEMENTS

We would to thank to all the people and organizations involved in the collection of photographic and sighting data over the years. In the Azores the online MONICET platform (http://www.monicet.net) had the collaboration of the whale-watching companies Terra Azul — Azores Islands Whales and Dolphins, Picos de Aventura — Animação e Lazer Lda., Azores Experiences - Whale Watching & Jeep Tours, Ocean Emotion Azores Whale Watching, SeaColors Expeditions and Futurismo —Azores WhaleWatching. In Madeira, we thank to whale-watching operators Ventura | nature emotions, H2O-Madeira, Seaborn, VMT Madeira and Lobosonda, and in the Canary Island, to whale-watching operator OCEANO Gomera.

### Funding

Partnership Program (US) and project "Cetáceos, Oceanografía y Biodiversidad de las Aguas Profundas de La Palma y El Hierro" funded by "Ministerio de Ciencia e Innovación" of the Spanish Government, grant number CETOBAPH-CGL2009-1311218 supported the work in the Canary Island. In Madeira, this study was supported by the Oceanic Observatory of Madeira through the project M1420-01-0142-FEDER-000001 and by the Portuguese Foundation for Science and Technology through the strategic project UID/MAR/04292/2020. Ana Dinis and Filipe Alves have grants funded by ARDITI— Madeira's Regional Agency for the Development of Research, Technology and Innovation, throughout the project M1420-09- 5369- FSE- 000002, and Annalisa Sambolino is supported by the Portuguese Foundation for Science and Technology through the PhD grant number SFRH/BD/1416092018. In Azores, the MONICET platform is supported by project MEEMO (ACORES-01-0145-FEDER-000079) and Marta Tobena is supported by a PhD grant (M31a/F/0722015). There was no additional external funding received for this study. The funders had no role in study design, data collection and analysis, decision to publish, or preparation of the manuscript.

### Grant Disclosures

The following grant information was disclosed by the authors:
"Ministerio de Ciencia e Innovación" of the Spanish Government: CETOBAPH-CGL2009-1311218.
ARDITI-Madeira's Regional Agency for the Development of Research, Technology and Innovation: M1420-01-0145-FEDER-000001-OOM and M1420-09-5369-FSE-000002.
Portuguese Foundation for Science and Technology: UID/MAR/04292/2020 and SFRH/BD/1416092018.
MEEMO: ACORES-01-0145-FEDER-000079.
PhD grant: M31a/F/0722015.

### Competing Interests

Sara Magalhães is employed by Mar Ilimitado, Rui Peres dos Santos is employed by Futurismo and Fabian Ritter is employed by M.E.E.R. e.V.

## Author Contributions

- Ana Dinis conceived and designed the experiments, performed the experiments, analyzed the data, prepared figures and/or tables, authored or reviewed drafts of the paper, and approved the final draft.
- Carlota Molina analyzed the data, prepared figures and/or tables, authored or reviewed drafts of the paper, and approved the final draft.
- Marta Tobeña, Annalisa Sambolino, Karin Hartman, Marc Fernandez, Sara Magalhães, Rui Peres dos Santos, Fabian Ritter, Vidal Martín, Natacha Aguilar de Soto and Filipe Alves performed the experiments, authored or reviewed drafts of the paper, and approved the final draft.

## Data Availability

Raw data is available in the Supplemental Files.

## Supplemental Information

Supplemental information for this article can be found online at http://dx.doi.org/10.7717/peerj.11069#supplemental-information.

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
