# Peer review of "Large-scale movements of common bottlenose dolphins in the Atlantic: dolphins with an international courtyard"

_PeerJ, doi:10.7717/peerj.11069_

## Round 0.1 · original submission · Major Revisions

I have now received two good reviews on your manuscript recently submitted. Both reviewers, who are specialists in the field, were very positive about your results. I also think the manuscript is in overall good shape and congratulate the authors on their effort to put together and analyze such a massive international dataset. However, they also pointed out a number of aspects that deserve the author's attention and could potentially improve the quality of the presentation. R2 was a bit more critical and listed several suggestions that need to be incorporated into the revised version. I kindly ask you to pay special attention to his/her comments regarding data on distance travelled, social analysis, and photograph quality.

From my view, the critical downside of the paper is the lack of proper quantitative analysis, for example, involving photo ID software (see PDF attached for details) and mark-recapture techniques based on photos. The only figure with analysis is the individual network, but no network metric was calculated. Thus, I really believe that authors must work a bit harder to improve the quantitative side of the manuscript, in addition to providing more details on the Methods.

In the Results, it's quite confusing the information on specific individuals betweeen years. Try to summarize it and point to the general patterns, instead. The Conclusion is pretty long too, this section usually summarizes the most interesting results only, and it's good practice to make it one paragraph only.

I have provided minor edits in the pdf attached.

PeerJ has good resources for how to write your Rebuttal letter https://peerj.com/benefits/academic-rebuttal-letters/

Reviewer 1 ·

Basic reporting

The manuscript provides important information of the movements of bottlenose dolphins in the NE Atlantic Ocean.
The article is well written with sufficient references, data and figures.
Grammatical changes are suggested in the "General comments for the author" section.

Experimental design

This study represents a well example of the importance of sharing information between research groups from different areas. It compiles information from three different countries, three autonomous regions and continental information, resulting in novel findings on the species movement patterns.

Validity of the findings

Even though the data used is sufficient, the results could be more robust if complemented with likelihood techniques as used by Alves et al 2019 for Short-finned pilot whales of Macaronesia.
However, the descriptive analysis provided in the manuscript has a significant contribution for the ecology of the species.

Additional comments

I suggest the authors to make the following grammatical changes to the manuscript:

L42: Replace “Canaries” with “Canary Islands” for consistency
L43: Replace “as well as among” with “neither between”
L60: The “s” is missing in “Wells”
L67: The “s” is missing in “Wells”
L69: Replace “Reynolds et al” with “Reynolds, Wells & Eide”
L89: Should be species’ instead of species, as it is a possessive noun
L111: Period “between” 2004 “and” 2016 instead of “period 2004 to 2016”
L115: Add “Islands” after the word Canary
L137: Replace “felled” with “grouped”
L178-180: This information is redundant since it is already included in the figure detail
L189: “and” is written in capital letters
L206: Replace “maintain” with “have”
L224: Replace “11” with the appropriate reference
L237-238: “This strongly […] Madeira” is redundant with the previous sentence
L242: Should be dolphins’ instead of dolphins, as it is a possessive noun
L250: Maybe include the reference “Louis et al 2014”
L251: Add “Islands” after the word Canary
L262: Replace “found” with “find”
L276: Replace “realized” with “performed”
L280: Remove the “s” in “accounts”
L282: Add an “s” to “dolphin”
L283: Replace “bycatch” with “by-catch” for consistency
L302: Replace “Canarias” with “Canary Islands” for consistency
L385: This reference is not included in the text
L451: Replace “trucatus” with “truncatus”
In figures 3, 4 and 5: Add an “s” to the word “dolphin”
Along the manuscript there are many times where there are commas preceding the word "and". I recommed not using commas before "and" when enumerating. For instance, "Azores, Canary islands and Madeira" instead of "Azores, Canary islands, and Madeira"

Reviewer 2 ·

Basic reporting

This manuscript provides useful information on the actual and potential connectivity among local populations of bottlenose dolphins in various Macaronesian island systems, and the Portuguese mainland. It is a nice example of collaboration among research groups, and nicely shows the mobility of these animals. In this regard, it is certainly a useful study worth to be published. I commend the authors on undertaking this important work.

However, I do have some concerns.

There are some unclear aspects and inconsistencies. Moreover, the authors place quite a bit of emphasis on the distances travelled in the Discussion, but they do not actually provide these distances in the Results (or elsewhere).

The treatment of existing literature is somewhat incomplete, as some relevant literature sources are missing from the manuscript.

In addition, the manuscript would benefit from proof-reading and revision for language.

I provide some specific comments below and encourage authors to carefully consider them when revising their manuscript.

Experimental design

Description of methods is rather vague and details are missing to get a good idea of what was done and why. There may be some important caveats (photographic quality), which may have significant influence on the results, but that do not appear to be addressed at all. Moreover, certain terminology is used either incorrectly or is at least a bit misleading – for example the “social analysis” is not truly a social analysis, and should therefore be framed a bit differently. This is not to say that the exercise of “mapping” known connections among individuals is not useful – it is – but it should be clearly identified as such, and not confused with social analysis, because that gives the impression of this being a social structure or a social network study.
These issues should be resolved before the manuscript is appropriate for publication.

Validity of the findings

As stated above, this manuscript provides useful information on the actual and potential connectivity among local populations of bottlenose dolphins in various Macaronesian island systems, and the Portuguese mainland. The results themselves are interesting and important enough.

However, in some places the authors infer or speculate beyond what the data show. I am not necessarily discouraging the authors to explore various implications, but they should "tone down" the implied certainty or relevance. More details in the specific comments.

As mentioned above, there are some methodological consideration which may affect results. These should either be incorporated in the study better, or at least clearly and carefully addressed in the discussion.

Additional comments

SPECIFIC COMMENTS

TITLE: My suggestion would be to drop the “novel insights” and just keep “Large-scale movements of common bottlenose dolphins in the NE Atlantic: dolphins with an international courtyard”.
This way the title is both shorter (and “cooler” in my view), without trying to force the “novel insights” into it. With “novel insights”, it sounds as if you are merely providing some new insights into an already studied thing. But instead, as far as I’m aware, this large-scale study is brand new – so remove the “novel insights” accordingly.

ABSTRACT
L39: Replace “photos” with “photographs”. Also, a space is missing between up and 1791.

L43: “as well as among the three archipelagos” – This is confusing. By the three archipelagos I presume you mean Azores, Madeira and Canaries? If so, the way it is written sounds as if no matches were made among those. But you just said a few lines earlier that there were matches between Azores and Madeira and Madeira and Canaries. So something doesn’t add up here.

L46: What are “all known residency patterns”? Please re-word of remove from the abstract. I understand it is probably explained better later on in the paper. However, the abstract should be almost as a “stand-alone” piece explaining the study, so it should be clear on its own.


INTRODUCTION

L60: Range implies overall distribution range of the species. You probably mean “ranging patterns” or “home range” of specific populations? Please re-word accordingly.

L61: Coastal, not costal.

L63: Oudejans et al. 2015 is not in reference list.


MATERIALS AND METHODS

L97: please replace “habitat” with “area”. Also, change “in” to “along” or “off” Iberian peninsula.

L102: Why is there an apostrophe in “cetacean” (cetacean’)...?

L107-108: “representing three different archipelagos, and a small area on the coastal Portuguese mainland” - This has been said already, please avoid unnecessary repetitions.

L112: “compared” – I suggest replacing to “used” or “included”, because the way it is written now, it makes it seems that those two catalogues were compared to each other (rather than to other catalogues as well).

L114-115: “A third set of data from this region, containing 201 photos of 42 individuals collected from 2014-2016 was added.”
Wait... does this still refer to Azores? If so, why make it confusing like this? Just say they were 3 catalogues from the beginning, rather than first saying there were two, and then saying another one was added... If there were three, then say it was three from the start.

L117: “from 2014” – I assume this means only in year 2014 (as opposed to from 2014 onwards)? To avoid confusion, remove “from” and just leave the year.

L128: Please explain what you mean by “ambiguous matches”.

L129: But not false negatives...?

L122-129: There is no mention of whether photographic quality was assessed and how. This has major implications on the results and relates back to the ambiguity and false positives (and false negatives). This is a major issue and if not addressed, may seriously bias the reported results. This should be either incorporated into the analysis, or at the very least incorporated into the caveats and interpretation of results in the Discussion. Poor photographic quality may create matches where there are none (false positives) or create absence of matches where there actually should be some (false negatives). Both of these may severely affect the outcomes of matching. This is something the authors need to consider.

L132-133: Please explain a bit why only Madeira animals were used for this. In other words, if the idea was to get an overall sense of connectivity among all putative local populations, then it would make sense to include all. On the other hand, if the idea was to only look at social connectivity among Madeira animals, then why was not the same done for other places such as Azores, etc.? It seems a bit arbitrary.

L137: Some revision of language and spelling needed here (“remain” individuals, “felled”,...).

L137-139: This statement is not logical. Interactions between dyads form part of the social interactions at the level of the community. These terms are not mutually exclusive. Even when looking at interactions at the level of community, the sighting frequency is still important, so the logic of the statement does not hold. However, I do understand what the likely aim was here: to get the basic idea of connectedness among individuals. If so, I suggest this is re-worded, so it accurately reflects that. Also, the sub-heading should be changed – this is not really “social analysis”, it is a crude estimate of connectedness (which is fine and useful). Therefore, please re-word accordingly, otherwise this gives the false impression that the authors carried out an actual social analysis, which is not the case.

L140-142: It is not entirely clear to me what was done or why. More details are needed here, with a better explanation. For example, since the paragraph starts with saying only Madeira animals were included, it is confusing where “with Macaronesian individuals” fit in here. If I am reading this correctly, the authors formed a network of Madeira animals and then added “Macaronesian individuals” (whatever this means – are these all other animals, or only those animals re-sighted between Madeira and other island systems...?), but this is not entirely clear. Also, unclear what “main clusters” refers to. Furthermore, “to highlight their associations” – again not sure what exactly is meant by this, because some of these associations can be misleading if animals seen only once were included (see previous comments about this).
So, more detail and clarity is needed here. It is all a bit vague at the moment.

L142: Please provide more detail about why and how this was included as an attribute in the analysis. I may try to guess this because I know what “attribute” means in the context of social analysis, but it is not immediately apparent, especially not to a general reader. Also, it is unclear to me how this attribute was incorporated into the analysis.
(re-reading this now, I realise you probably meant how this is labelled in Figure 6 – in this case, you can remove this sentence here altogether, because Figure 6 already provides that information)


RESULTS

L148: Some revision of language needed.

L148: Please be specific. What does “most of which” mean? Please say exactly how many matches between Madeira and Canaries, and how many between other places. It is too vague.
Also, be specific in both the number of matches but also number of animals.

L149: “Additionally, three individuals were seen both in the Azores and the Madeira archipelagos”
Additionally to what? To those 26 matches? Or is this part of the 26? It is not clear, because previous sentence says MOST of 26 matches were between Madeira and Canaries (but not all). So, are these three individuals part of those remaining matches among 26, or in addition to that?

L151: But what about Canaries-Sagres and Azores-Sagres...? Why are these not included in the comparisons?

L166: What do you mean by “possibly accompanied by”? Were they seen together or not? If not, do not speculate beyond data. Or if speculation is warranted, leave it for Discussion, not Results.

L173: Please provide movement distances for all animals.

L178: “...including 20 Macaronesian individuals seen in association...” – What does this mean?

L178: Which diagram? You should refer to specific figures.

L180-183: Some sentence re-wording is needed for proper language structure.

L182: I may be missing something, but from Fig 6, it does not show that animals seen in both Azores and Madeira associated with “migrants”, because no lines goes from them to other migrant (yellow) animals.



DISCUSSION

L186-187: This is demonstrably not true, because the authors state in the Results (and Figures) that no animal was seen in all three archipelagos. And yet, now the authors state that 26 dolphins moved between three archipelagos. Please clarify the sentence to accurately reflect which is the case.

L189: Why is “AND” capitalised?

L193-195: This is a rather incomplete list. See some more examples below, and incorporate them into the manuscript:
- Gnone, G., Bellingeri, M., Dhermain, F., Dupraz, F., Nuti, S., Bedocchi, D., ... & Azzellino, A. (2011). Distribution, abundance, and movements of the bottlenose dolphin (Tursiops truncatus) in the Pelagos Sanctuary MPA (north‐west Mediterranean Sea). Aquatic Conservation: Marine and Freshwater Ecosystems, 21(4), 372-388.
- Hwang, A., Defran, R. H., Bearzi, M., Maldini, D., Saylan, C. A., Lang, A. R., ... & Weller, D. W. (2014). Coastal range and movements of common bottlenose dolphins off California and Baja California, Mexico. Bulletin, Southern California Academy of Sciences, 113(1), 1-13.
- Defran, R. H., Weller, D. W., Kelly, D. L., & Espinosa, M. A. (1999). Range characteristics of Pacific coast bottlenose dolphins (Tursiops truncatus) in the Southern California Bight. Marine Mammal Science, 15(2), 381-393.

Also, albeit shorter distances, but also possibly useful to consider in the context of bottlenose dolphin photo-ID and movements, may be:
- Bearzi, G., Bonizzoni, S., & Gonzalvo, J. (2011). Mid-distance movements of common bottlenose dolphins in the coastal waters of Greece. Journal of Ethology, 29(2), 369-374.
- Genov, T., Angelini, V., Hace, A., Palmisano, G., Petelin, B., Malacic, V., ... & Mazzariol, S. (2016). Mid-distance re-sighting of a common bottlenose dolphin in the northern Adriatic Sea: insight into regional movement patterns. Marine Biological Association of the United Kingdom. Journal of the Marine Biological Association of the United Kingdom, 96(4), 909.

L196-197: “The distances reported here for the individuals that moved between Madeira and the Canary Islands are comparable...” – The authors have not provided any distances in the Results, and one cannot readily infer that from the maps. Please provide actual movement distances for all individuals seen in different archipelagos.

L198-199: Impossible to assess this statement, because there are no distances reported.

L207-210: This statement is a bit vague and not very informative. What was the aim here, what is the point the authors are making?

L207-208: You may want to consider the following (more recent) relevant publication on this topic: Bräger and Bräger 2019: Movement Patterns of Odontocetes Through Space and Time. In: B. Würsig (ed.), Ethology and Behavioral Ecology of Odontocetes, Ethology and Behavioral Ecology of Marine Mammals, https://doi.org/10.1007/978-3-030-16663-2_6.

L214: comparably

L221: Please add the species name here, because Shark Bay dolphins are not the same species.

L222-223: “will be improved”, “will decrease” – How can you be so sure? “Will” implies certainty. Consider replacing with “may” or “could”.

L224: What is “(11)”? This should be a citation, I presume.

L242-243: Also see the following in relation to this, showing marked within-population variability, and making the point of variability among populations:
Genov, T., Centrih, T., Kotnjek, P., & Hace, A. (2019). Behavioural and temporal partitioning of dolphin social groups in the northern Adriatic Sea. Marine biology, 166(1), 11.

L247-250: These results do not show that. I’m afraid the authors are making inferences beyond the available data. Yes, the authors can provide this speculation as one way of interpreting results, but please “tone down” the certainty related to this. You may certainly refer to it as a possibility, but please do not say that results show this, because they do not.

L262: Some revision of English is needed (find, found).

L269-270: This is very vague – please provide more rationale.


CONCLUSIONS

L273: Does it? I thought so too, but the authors themselves report the study of Tobeña, so apparently this study does not provide first evidence of this. Please remove “first”. It is still relevant enough without it.

L275-276: Please provide references.

L282-285: Some references would be good here.

L297: What do you mean “shall be effective”? Of course they should be effective, but not clear what this statement aims to say.

L307-308: Please provide the justification why this would be the most effective way. I am not saying a sanctuary is not a good way forward, but I am not convinced it is necessarily the most effective way. For example, you may have no MPA at all, but the regulation of issues/threats (fisheries, for example, or dolphin-watching) may actually provide greater benefit to the animals – which does not need to be in the form of an MPA necessarily.
This statement is too generic and not supported. So either provide a better justification (and some references), or remove it.


FIGURES: Please add movement distances into Figures 3-5.

FIGURE 6: The additional levels of symbols, and the description and legend of location is not needed, because it location is already depicted by squares and the names of the locations, so the extra legend is a bit obsolete.

---

## Round 0.2 · Major Revisions

I couldn't get the same reviewers to comment on this revised version. I carefully checked the response to reviewers and the revised version of the text. I'm sending out this decision with a major revision because, although authors made a huge effort to address all questions raised in the previous round of review, I agree that some aspects still need to be improved and clarified. I don't think you'll have much work to incorporate these last amendments to the text.

·

Basic reporting

no comment

Experimental design

It’s not stated but did the researcher review each photo/fin from these separate catalogues for quality and distinctiveness? I understand that each institution/researcher will have its own way of procedures including grading the photographs for quality (focus, contrast etc.) and for dolphin dorsal fin distinctiveness but it is well known that various researchers don’t see quality and distinctiveness the same (Urian et al. 2015) which can add to false positives/negatives. To avoid the ambiguity of grading between different researchers/intuitions and different catalogues I suggest the researcher reviewing the photos for the greater Macronesia catalogue should separately grade for distinctiveness and quality for each photo/dolphin when reviewing them for that catalogue. It then could be reported on how many photos/individuals were removed due to low quality/distinctiveness if indeed any were.

It’s reported that matches were confirmed by a second researcher, but I am more concerned with false negatives (missed match – recording one animal as two). A second experienced research should review the entire catalogue for false negatives. Even better would be to use a newer computer assisted matching program. I am not familiar with Darwin, (and it seems that only one catalogue was used with Darwin) but comparing 1250 individual dolphins by naked eye is a huge undertaking which assumes some risk of human error. Using a program such as FinFindR (https://github.com/haimeh/finFindR) or Flukebook would help a great deal with matching in a minimal amount of time (Thompson et. al 2019, Levenson et al. 2015). I’d hate to see that they have missed a few matches that could add to the robustness of this manuscript due to human error.

Levenson, J., S. Gero, J. Van Oast, and J. Holmberg. 2015. Flukebook: a cloud-based photo-identification analysis tools for marine mammal research. Accessible at: https://www.flukebook.org

Thompson, J.W., Zero, V.H., Schwacke, L.H., Speakman, T.R., Quigley, B.M., Morey, J.S. and McDonald, T.L., 2019. finFindR: Computer-assisted Recognition and Identification of Bottlenose Dolphin Photos in R. bioRxiv, p.825661.

Urian, K., Gorgone, A., Read, A., Balmer, B., Wells, R.S., Berggren, P., Durban, J., Eguchi, T., Rayment, W. and Hammond, P.S., 2015. Recommendations for photo‐identification methods used in capture‐recapture models with cetaceans. Marine Mammal Science, 31(1), pp.298-321.

Validity of the findings

no comment

Additional comments

This manuscript presents new movement information on the common bottlenose dolphin in this area. I commend the authors for this important contribution which would add to the understanding of common bottlenose dolphins in the northeast Atlantic. However, there are a few issues stated in the "Experimental Design" section and in line specific comments that could be addressed that would strengthen the manuscript. I hope you find them useful.

Line specific comments:

TITILE: NE should be written out.

ABSTRACT:
L30: insert “common” before bottlenose.

L31: replace NE with “North East (NE)”

L33: the K in “km” is not capitalized and further along in the manuscript it is. Be consistent.

L57: change “threats” to “anthropogenic disturbances, such as…”

L102-119: Would the numbers from the catalogues be more appropriate in the results rather than the methods? I go back and forth on that.

L105: Please define the acronyms the first time in use. Replace “OOM” with “Oceanic Observatory of Madeira (OOM)”

L110: define “MONICET”

L115: define “SECAC”

L117: define “BIOECOMAC” and “MEER”

L121-141:Photo-identification Procedures – While I appreciate the changes in the methods, I would like to see a bit more clarification. Is the description of the photo-identification procedures from the individual catalogues? Or is it describing the methods that were used to create the greater Macronesia catalogue? It wasn’t very apparent at first. A bit of editing would improve the manuscript and give more clarification.

L138: Darwin 2.0 needs a reference.

L144: For clarity after “was compiled” insert “from the individual catalogues”

L148: “two or more locations were included…” Does this mean resighted two or more times or seen in two different archipelago or two island areas within each archipelago?

L148: “100% certainty” Were the photos also confirmed with a second experienced researcher?

L151-162 Associations. I understand why the authors have included this, but I am not sure how much this adds to the manuscript without the use of more rigorous analysis. There isn’t much detail on how they manipulated the data for use in Netdraw. SOCPROG could be used to help with this. But there could be an issue on how the data was collected (from various researchers and different platforms) that the analysis could not be done in the first place.

L175: remove “.” after “Islands”

L229-230: Does the Azores archipelago not share these biogeographic and oceanographic characteristic? Please explain rather than just give the reference. Shorter distance between the two archipelagos seems to be the reason for the matches. Flip the sentences and expand on your reasoning.

L278: remove the parentheses around the word “much”

Figures 3-5: The exact sighting locations would be helpful. Instead of a generic dolphin photo, perhaps photos the dorsal fins of the animals represented would be better.

---

## Round 0.3 · accepted · Accept

I think the last additions and corrections made by the authors improved the paper and I'm glad to recommend it for publication. Congratulations.